# Role of Exosomes in the Pathogenesis and Theranostic of Alzheimer’s Disease and Parkinson’s Disease

**DOI:** 10.3390/ijms241311054

**Published:** 2023-07-04

**Authors:** Aojie He, Meiling Wang, Xiaowan Li, Hong Chen, Kahleong Lim, Li Lu, Chengwu Zhang

**Affiliations:** 1School of Basic Medical Sciences, Shanxi Medical University, 56 Xinjiannan Road, Taiyuan 030001, China; heaojie@sxmu.edu.cn (A.H.); wangmeiling@sxmu.edu.cn (M.W.); lxw@b.sxmu.edu.cn (X.L.); chenhong22@sxmu.edu.cn (H.C.); 2Lee Kong Chian School of Medicine, Nanyang Technological University, 11 Mandalay Road, Singapore 308232, Singapore

**Keywords:** exosomes, Alzheimer’s disease, Parkinson’s disease, pathogenesis, theranostic

## Abstract

Alzheimer’s disease (AD) and Parkinson’s disease (PD) are the most common neurodegenerative diseases (NDDs) threatening the lives of millions of people worldwide, including especially elderly people. Currently, due to the lack of a timely diagnosis and proper intervention strategy, AD and PD largely remain incurable. Innovative diagnosis and therapy are highly desired. Exosomes are small vesicles that are present in various bodily fluids, which contain proteins, nucleic acids, and active biomolecules, and which play a crucial role especially in intercellular communication. In recent years, the role of exosomes in the pathogenesis, early diagnosis, and treatment of diseases has attracted ascending attention. However, the exact role of exosomes in the pathogenesis and theragnostic of AD and PD has not been fully illustrated. In the present review, we first introduce the biogenesis, components, uptake, and function of exosomes. Then we elaborate on the involvement of exosomes in the pathogenesis of AD and PD. Moreover, the application of exosomes in the diagnosis and therapeutics of AD and PD is also summarized and discussed. Additionally, exosomes serving as drug carriers to deliver medications to the central nervous system are specifically addressed. The potential role of exosomes in AD and PD is explored, discussing their applications in diagnosis and treatment, as well as their current limitations. Given the limitation in the application of exosomes, we also propose future perspectives for better utilizing exosomes in NDDs. Hopefully, it would pave ways for expanding the biological applications of exosomes in fundamental research as well as theranostics of NDDs.

## 1. Introduction

AD and PD are the two most common neurodegenerative diseases, for which the etiology remains to be elucidated. AD and PD share comparable pathological features, such as the progressive loss of specific neurons, and the presence of aggregated proteins [1,2]. The exact mechanisms underlying the pathogenesis of AD and PD are not yet fully understood. Mitochondrial dysfunction, oxidative stress, and compromised protein degradation and inflammation are all involved in the pathogenesis of both AD and PD. There are also distinct participants for each disease. Regarding AD, the Aβ/tau hypothesis is widely accepted, and accumulation of Aβ/tau is thought to be the initiator as well as the biomarker of AD. Cerebrovascular dysfunction is also reported to participate in the development of AD [3]. In terms of PD, it is believed that the aberrant accumulation of α-synuclein (α-syn) is the key mediator of the apoptosis of dopaminergic neurons. Pesticides are one environmental factor that could potentially lead to PD [4,5]. In recent years, with the prevalent usage of electronics such as cellphones, the impact of microwaves on neurodegeneration has also attracted the attention of researchers. Sohail Mumtaz and his colleagues addressed this issue and reported that specific frequencies of microwave radiation can potentially lead to hippocampal damage and impair cognitive function [6]. Furthermore, when it comes to explaining the mechanisms behind neurodegenerative diseases, there exists a relatively innovative idea suggesting that the manifestation of AD and PD is a loss of memory and learning ability and a movement disorder, respectively [1,2]. So far, AD and PD remain incurable. In the clinic, the diagnosis of AD and PD predominantly depends on clinical manifestations together with bioimaging such as computed tomography (CT) and magnetic resonance imaging (MRI) [7,8,9]. Treatment of AD and PD mainly involves symptomatic approaches, rather than approaches aimed at terminating or rescuing pathological processes involved in neurodegeneration. Inhibitors of acetyl-cholinesterase (AChE), including donepezil, rivastigmine, and galantamine, are first-class medications for AD, which can temporarily or partially restore memory but not reverse the loss of neurons [10,11]. Administration of L-DOPA, catechol-O-methyltransferase (COMT) inhibitors, monoamine oxidase inhibitors, amantadine, and dopamine agonists can effectively relieve the symptoms of PD, but it barely delays the loss of dopaminergic neurons [12,13]. The onset of AD and PD is latent. When patients visit a doctor for treatment, the loss of neurons has occurred for decades, and intervention hardly achieves anticipated effects. Accordingly, novel strategies for the diagnosis and treatment of AD and PD are in high demand.

Exosomes are microvesicles with a diameter of about 40–150 nm secreted by various types of cells such as neurons, immune cells, and mesenchymal stem cells (MSCs), which function as the “cargo” for intercellular materials and information transferring [14,15,16]. In 1983, exosomes were first identified in sheep reticulocytes, and in 1987 Johnstone et al. named them “exosomes” [17,18]. Accumulating studies reveal that exosomes possess more functions than deposing metabolites. Exosomes contain proteins, lipids, and nucleic acids, which can be released into the extracellular space and transferred to peripheral tissues [14]. One crucial biological basis for exosomes to fulfill their function is their ability to smoothly traverse various physiological barriers, including cell membranes and the blood-brain barrier (BBB). The lipid bilayer membranes of exosomes facilitate its fusion with membrane-like structures in the body such as the BBB [19]. Once exosomes enter neighboring or distal cells, the “cargo” released by exosomes alter the status and functions of those cells [20,21]. It is widely acknowledged that the status of the immune system and communication among immune cells play vital roles in the development of diseases. Exosomes are believed to participate in immune regulation. Exosomes present in the bloodstream facilitate the transmission of inflammatory cytokines, chemokines, transcription factors, and proteins from the stressed microenvironment to the brain, and activate microglial cells and astrocytes, which subsequently lead to the occurrence of NDDs [22]. Moreover, the “cargo” in the exosomes can also serve as biomarkers for diseases [23]. In recent years, exosomes have been engineered to endow them with more functions. However, the role of exosomes in the pathogenesis and therapeutics of NDDs has not yet been elaborately addressed.

## 2. Exosomes

Exosomes were initially thought of as one way for cells to dispose of waste, trash, and unneeded cellular components [24]. Yet with the progress of intensive research, more functions of exosomes have been discovered. Valadi et al. reported that exosomes contain messenger RNAs (mRNAs) and non-coding RNAs (e.g., miRNAs), which might be one new mechanism for the exchange of genetic material between cells [25]. Moreover, other researchers demonstrated that exosomes carry not only RNA but also proteins, lipids, and other biomolecules, which play an essential role in intercellular material and information transmitting. The presence of “cargo” within exosomes underlies its potential utility as diagnostic biomarkers. Furthermore, exosomes have been discovered to harbor a diverse range of organelles, such as mitochondria, components of the endoplasmic reticulum, and fragments of the Golgi apparatus, further amplifying the function of exosomes [26]. Ljubava D. Zorova et al. utilized real-time quantitative PCR to verify the existence of mitochondria DNA (mtDNA) in exosomes [27]. Xiaowan Wang et al. proved that exosomes from NT2 cells also contain mtDNA, which aligned with the finding of Ljubava D. Zorova’s group [27,28]. The presence of mitochondria in exosomes awaits support from more studies.

Exosomes can be found in diverse bodily fluids, such as blood, cerebrospinal fluid (CSF), sweat, urine, and saliva. Given its wide presence and transference in body fluids, exosomes exhibit promising applications in the diagnosis and therapeutics of various diseases [29,30]. It should be noted that exosomes belong to a family of extracellular vesicles (EVs). EVs could be classified into different subtypes based on their diameter and biogenesis as shown here in Table 1 [31,32]. 

### 2.1. Biogenesis and Composition of Exosomes

Exosomes are sequentially formed via the endosomal pathway (Figure 1), which includes the early sorting endosome (ESE), late sorting endosome (LSE), and finally, the multivesicular bodies (MVBs) that fuse with the cell membrane and are released [14]. Exosomes have a lipid bilayer membrane consisting of glycerol diacids, phospholipids, glycerophospholipids, polyglycerophospholipids, cholesterol, and sphingolipids, which are more rigid than the plasma membrane, ensuring the stability of exosomes in the external environment [14,33]. 

The components of exosomes determine their biofunction. Exosomes are first thought to be “garbage bags” for cells to get rid of useless cellular components [24]. Recent studies reveal that the “cargo” carried by exosomes possess multiple biological functions [29,33]. The “cargo” includes proteins, metabolic enzymes, and nucleic acids. To be noted, the “cargo” in exosomes might be diverse depending on the cell types from which they are derived [29,34]. The featured exosomal proteins are a four-transmembrane protein superfamily, including CD63, CD9, CD81, and CD82, which were widely utilized to identify exosomes [35,36]. However, Mathilde Mathieu et al. pointed out that CD63 is an exosome-specific protein, while CD9 and CD81 might not be [37]. There are other functional proteins in exosomes, such as the endosomal sorting complex-related proteins required for transport (e.g., Tsg101), lysosome-associated membrane glycoproteins (e.g., LAMP-1 and LAMP-2B), multivesicular body-associated proteins (e.g., Alix-1), heat shock proteins (e.g., hsp60, hsp79, and hsp90), adhesion molecules (e.g., CD45 and CD11b), histocompatibility-related MHC-I and II, Rabs family proteins, and membrane-linked proteins and integrins [38]. The enzymes carried by exosomes mainly include GTPase and metabolic enzymes [14,16]. Alongside proteins, exosomes also contain nucleic acids such as mRNAs, non-coding RNAs, long non-coding RNAs, and circular RNAs that help to transmit genetic information and influence the function of target cells [14]. It was reported that exosomes from living cells and dying tumor cells could release DNA. Nevertheless, the presence of DNA in exosomes is still controversial, and further study is needed [39,40]. Omics on exosomes could help to define their exact “cargo”, and more novel components might be found.

### 2.2. Exosomes Uptaken by Target Cells

Exosomes play an essential role in intercellular information exchanges by delivering bioactive molecules to the receptor cells, which in turn affects the physiological processes and disease development of the organism. Exosomes are uptaken by adjacent recipient cells after being released into the extracellular space by the donor cells [20,21]. The modes of exosomes uptaken by receptor cells are classified into three pathways: (1) Endocytosis: Endocytosis is one of the dominant manners by which exosomes enter recipient cells, which could be mediated by clathrin-dependent or clathrin-independent pathways, lipid rafts-mediate internalization, micropinocytosis, and phagocytosis [41]. (2) Direct fusion: Exosome membrane fuses with the plasma membrane of the recipient cell, and then “cargo” such as exosomal miRNAs are released into the cytoplasm of the recipient cell [14]. (3) Binding exosomal surface proteins: Exosomes bond to the receptors on recipient cell membranes such as the integrin-quadruple transmembrane protein complex and then are transferred into cells [42].

### 2.3. Functions of Exosomes

Exosomes were initially thought to be one way for cells to eliminate waste products, and to remove excess or unnecessary substances from cells so as to maintain the homeostasis of the intracellular environment [24]. Later on, it was found that exosomes could exchange materials and information between cells, and impact the physiopathological activities of target cells [34]. Exosomes can not only act on targeted cells by paracrine secretion, but also impact the peripheral or distal targets by circulation [15]. In recent years, exosomes derived from MSC exhibited satisfactory outcomes in the treatment of diseases. Xin et al. reported that MSCs-derived exosomes promoted neurites outgrowth and the functional recovery of a rat stroke model [43]. Exosomes exhibited comparable effects to those of MSCs, but it steered clear of the limitation of stem cell transplantation such as immune responses, tumorigenesis, and inadequate differentiation [44]. Based on the biological properties of exosomes, such as their small size, good biocompatibility, prolonged circulation time in fluids, and ability to penetrate deep tissues, they have been developed as carriers for drug delivery [45]. Moreover, the “cargo” of exosomes is also closely correlated with the occurrence and progress of diseases, so the components of exosomes could serve as biomarkers for the diagnosis of diseases [23,46]. A notable observation is that exosomes are heterogenous. Exosomes secreted by different cells display distinct functions, and exosomes secreted by the same kind of cells also show diversity depending on the status of the cells. With the deepening of their study, more functions of exosomes will be revealed. 

## 3. Exosomes and AD

AD is the most common neurodegenerative disease. It is estimated that by 2050, the number of AD patients will reach 131.5 million, which poses a heavy global social and economic burden [47]. AD remains incurable, and AD patients inevitably suffer irreversible brain damage. Diagnostic and therapeutic methods face great challenges, and novel strategies are in high demand. Exosomes have opened up an alternative window for understanding pathogenesis and its theranostic [48].

### 3.1. Exosomes and AD Pathogenesis

AD occurs mainly due to the aggregation of misfolded proteins, resulting in neuronal degeneration in multiple regions of the brain, particularly the hippocampus [2]. AD is characterized by the presence of abnormally aggregated proteins. The aggregated proteins are mainly composed of amyloid-beta (Aβ), neurofibrillary tangles (NFTs), and the hyperphosphorylated Tau protein [2,10]. The underlying mechanisms of AD pathogenesis remain to be further elucidated. In recent years, the role of exosomes in AD pathogenesis has been receiving increasing attention.

Zheng et al. by tracking exosomes injected into AD mice found that plasma-derived exosomes accumulated in Aβ plaques, suggesting that exosomes participated in plaques formation [49]. Exosomes have been shown to transport Aβ and Tau proteins from damaged neurons to healthy ones, contributing to the formation of Aβ plaques and neurofibrillary tangles (NFTs) [50]. Exosomes released by microglial cells also contribute to the progression of AD. It is widely accepted that the activation of the PINK1/Parkin enhances mitophagy. When exosomes derived from M2 microglia are administrated into a cellular AD model (HT-22), mitochondria clearance is strengthened, and intracellular reactive oxygen species are decreased [51]. It is notable how the effects of exosomes can vary depending on the cell type from which they originate. Research has shown that exosomes derived from M1 microglial cells can activate quiescent microglial cells towards M1 polarization, leading to the release of pro-inflammatory factors and exacerbating pathological processes [52]. In AD mice, it has been revealed that Aβ is firstly translocated to MVBs and then to exosomes before being secreted out of neurons and transmitted [53]. Exosomes were also involved in tau protein transport in an AD mice model, and inhibiting exosomes synthesis could prevent tau protein dissemination in the brain [54]. Alongside proteins, miRNAs in exosomes derived from both CSF and blood of AD patients were found to impact Aβ genesis and accumulation [55]. Ding et al. reported that in the physiological state, *miR-185-5p* could target the 3′-UTR of an amyloid precursor protein (APP) transcript in N2a cells and transport it to receptor cells through exosomes, preventing the transcription of APP. In APP-overexpressing N2a, exosomes loaded *miR-185-5p* was reduced, thereby aggravating the pathological progress of AD [56]. Collectively, exosomes get involved in AD pathogenesis by transmitting materials or information between neurons as well as neuron-glia, which further leads to abnormal protein accumulation (e.g., Aβ, tau) and ultimate AD pathogenesis.

Collectively, AD is characterized by the aggregation of misfolded proteins, including Aβ and hyperphosphorylated Tau, leading to neuronal degeneration, especially in the hippocampus. Exosomes play a significant role in AD pathogenesis by participating in plaque formation, transporting Aβ and Tau proteins, and impacting Aβ genesis and accumulation through miRNAs. These findings highlight the involvement of exosomes in facilitating material and information transfer between neurons and neuron-glia interactions, ultimately contributing to abnormal protein accumulation and the progression of AD.

### 3.2. Exosome-Based AD Diagnosis

Currently, diagnosis of AD is mainly based on positron emission tomography (PET)/CT, cognitive behavioral syndrome (CBS), and biomarkers of AD pathology (Aβ1-42/1-40, T-Tau, p-Tau) [9,57]. However, bioimaging (PET/CT) and CBS-based diagnosis are often delayed due to the latent onset of AD. Biomarkers for checkup especially with CSF were invasive which brought injury to patients [58]. There is still a lack of methods to accurately predict or diagnose AD. In the clinic, the diagnosis of AD is multimode, including bioimaging, biochemical analysis, and questionnaires. Bioimaging such as PET or CT is costly, and the results could be interfered with by other kinds of dementia diseases [9]. Questionnaires are subjective and easily affected by the surveyor. Exosomes derived from neurons have characteristic receptors associated with nervous tissues, which may include neuron adhesion molecules and neurotransmitter receptors. Those receptors play a vital role in mediating the interactions between exosomes and target cells [59]. They facilitate the selective binding and uptake of exosomes, enabling the delivery of their “cargo” to specific cellular recipients. The presence of those receptors on exosomes facilitates their diagnostic application in NDDS [59,60]. Exosomes from patients with AD can be isolated from various biological fluids, including blood, urine, and saliva [60]. Hence, the non-invasive and convenient collection of exosomes, along with their stability after sample acquisition, further supports their utility in the field of diagnostics for AD and related disorders.

Ruihua Sun et al. using transmission electron microscopy (TEM) and nanoparticle tracking analysis (NTA) demonstrated that exosomes derived from the blood of AD patients were smaller in size and quantity compared to those from healthy controls [61]. Consistent with that finding, Antonio Longobardi et al. found that the number of exosomes in the blood of AD patients was 40% less than that of healthy controls. However, another group reported that exosomes from AD patients were larger than those of healthy controls [62]. Currently, there is a lack of consistentaffectede supporting the exact differences in exosome size between AD patients and healthy controls. The variation in exosomes size could be affected by various factors, including sample sources, collection techniques, and analysis methods. Further studies are needed to validate these differences and gain a deeper understanding of their role in the pathogenesis of AD and their potential diagnostic value [63]. Morphology of exosomes could be used as one parameter for diagnosis of AD, but the procedure for collecting and analyzing exosomes needs to be standardized. 

Proteins are important components of exosomes. In the exosomes derived from AD patients, β-site APP cleaving enzyme 1 (BACE-1), soluble peptide APP beta (sAPPβ), soluble peptide APP alpha (sAPPα), γ-secretase, and Aβ1-42 were found, which closely correlates with the pathogenesis and progression of AD [64]. Lipids of exosomes could also be used as promising biomarkers for AD diagnosis. With the help of semi-quantitative mass-spectrometry, Su et al. found that plasmalogen glycerophosphoethanolamine (PE) molecules (p-36:2, p-38:4) and lipids on the membranes of AD patients’ brain-derived exosomes were significantly upregulated compared with those of control groups [65]. 

MiRNAs, which are another kind of “cargo” from exosomes, have gained increased attention due to their role in the control of gene expression. It was proved that miRNAs in exosomes derived from AD patients showed significant alterations compared to those from healthy controls [66,67]. Liu et al. reported that nineteen miRNAs (e.g., *miR-15a-5p*) were upregulated, while five other miRNAs (e.g., *miR-15b-3p*) were downregulated in the exosomes from AD patients’ serum [68]. Gamez-Valero et al. checked the expression levels of miRNA in CSF of AD patients by microarray analysis, and found increased levels of *miR-132-5p*, *miR-485-5p*, and *miR-125b-5p*, as well as decreased levels of *miR-16-2*, *miR-29c*, and *miR-331-5p* [67]. MiRNAs in exosomes extracted from different body fluids, such as serum, plasma, and CSF, showed a distinguished profile, which should be considered when analyzing miRNA in AD and control groups.

The alteration in the “cargo” and levels of exosomes-derived biomarkers indicate their high potential value for AD diagnosis. Exosomes derived from a variety of body fluids ensures their availability and accessibility in diagnosis and exosomes derived from blood and neurons show even better creditability than CSF biomarkers or PET/CT.

### 3.3. Exosomes-Based AD Therapeutic

In the clinic, AD patients are usually treated with cholinesterase inhibitors, such as donepezil, galantamine, and rivastigmine, which are the first-line drugs [10]. However, these drugs can only relieve the clinical symptoms, rather than prevent neuronal death [11]. So far, AD is still incurable. In recent years, exosomes, as one novel intervention for AD, are attracting more attention due to their capacity to transfer bioactive substances and pass across the BBB [69,70]. It is predicted that exosomes could potentially represent an alternative strategy for “decellularization” AD therapy.

Hirohide Asai et al. found that exosomes secreted by microglia could disseminate tau, and the inhibition of exosome secretion reduces tau levels in mouse brains [71,72]. Exosomes biogenesis inhibition via the blocking of neutral sphingomyelinase 2 (nSmase2) activity, a critical enzyme regulating ceramide biogenesis, significantly decreased Aβ plaques as well as tau propagation in an AD mice model [73]. Those studies suggest that interfering with the formation of disease-associated exosomes could mitigate the progression of AD. There were also studies showing that exosomes exert neuroprotective effects in AD. Exosomes derived from curcumin-treated cells inhibited the phosphorylation of Tau protein by activating the AKT/GSK-3β pathway, thereby preventing neuronal death both in vitro and in vivo and alleviating the symptoms of AD [74]. It has been reported that *miRNA-193b* present in exosomes derived from the serum, plasma, or CSF of healthy donors could bind to the 3′-UTR region of APP and suppress the expression of APP [60]. Vella et al. reported that enzymes in exosomes such as neprilysin and insulin-degrading enzymes present in exosomes could degrade Aβ peptides and reduce extracellular and intracellular Aβ levels [75]. In addition, exosomes derived from human CSF or N2a cells improved the synaptic-plasticity-disrupting activity of both synthetic and AD cerebral-derived Aβ in vivo [76]. Those reports suggested that exogenous exosomes from healthy donors and cell lines ameliorated Aβ accumulation and delayed the progress of AD.

In recent years, exosomes derived from stem cells showed promising therapeutic effects in AD. Allaura S. Cone et al. demonstrated that exosomes derived from human umbilical cord-derived mesenchymal stem cells (hucMSCs) inhibited neuroinflammation and promoted the degradation of Aβ, thereby reducing Aβ aggregation in the brain of AD mice [77]. Xinyi Ma et al. demonstrated that exosomes secreted by adipose-derived mesenchymal stem cells (ADSCs) rapidly and efficiently entered the brains and accumulated mainly in the neurons of AD mice after intranasal administration Administering exosomes reduced Aβ deposition and decreased microglia activation, demonstrating that ADSCs-derived exosomes might serve as a feasible way for AD treatment [78]. Although stem cell-derived exosomes play an important role in AD therapy, there are difficulties in applying exosomes to clinical applications due to their limitations such as short half-life and poor targeting [79].

Given its ability to cross the BBB and carry “cargo”, exosomes have been engineered into the carrier to deliver therapeutic agents for AD treatment. Exosomes carrying beta-secretase (BACE1) siRNAs, intravenously injected into AD mice, alleviated disease-associated phenotypes by reducing Aβ levels [80]. Exosomes containing medicinal RNA, peptides, and synthetic drugs were applied in AD treatment and displayed optimistic results. Exosomes-mediated delivery of plant-derived bioactive components also showed a rescue effect in an AD model [81,82]. It is notable how exosomes tended to be trapped in peripheral tissues (especially liver and lung), leading to insufficient delivery to the brain. Appropriate modifications of exosomes could improve its delivery efficiency, targeting ability, and therapeutic efficacy [83]. Alvarez-Erviti et al. modified exosomes with rabies virus glycoprotein (RVG) and realized specific targeting of the brain [80]. Kim et al. modified exosomes with one peptide that enabled its binding to neurons, and nasal delivery of modified exosomes reduced the tumor necrosis factor- cell-alpha (TNF-α) level and cell apoptosis [84]. Engineered exosomes emerge as a new therapeutic strategy and are expected to be an alternative option for exosomes-based therapy [85]. In conclusion, there is significant potential in the utilization of exosomes as a drug delivery system for the treatment of AD (Figure 2).

## 4. Exosomes and PD

PD is the second most prevalent NDDS, significantly threatening the well-being of individuals, and of elderly people in particular [2]. The pathological feature of PD is the presence of intracellular aggregates that consist of α-syn [86]. The etiology of PD is thought to include age, genetic background, and environmental factors, which lead to the death of dopaminergic neurons of the substantia nigra of the midbrain by inducing mitochondria dysfunction, oxidative stress, and protein aggregation [87,88]. Earlier diagnosis and effective treatment of PD remain unmet. Therefore, defining novel biomarkers for preclinical diagnosis and exploring better strategies for the treatment of PD is a highly demanding pursuit.

### 4.1. Exosomes and Pathogenesis of PD

It is generally agreed that exosomes play a crucial role in the pathogenesis of PD, and the accumulation of α-syn is a feature of PD. Shi et al. injected ^125^I-labeled α-syn intracerebroventricularly (ICV) into the brains of mice and found that ^125^I-labeled α-syn was presented in blood exosomes, demonstrating that exosomes may serve as a α-syn disseminator to aggravate PD pathogenesis [89]. In addition, Emmanouilidou et al. found that α-syn was encapsulated in exosomes in SH-SY5Y and exported by a calcium-dependent endosomal mechanism, which illustrated the relationship between α-syn and exosomes from another perspective [90]. Gustafsson et al. reported that exosomes encapsulation could affect the uptake efficiency of α-syn in SH-SY5Y, but the mechanism had not been clearly elucidated [91]. In addition to delivering α-syn as “cargo”, exosomes could also serve as an incubator for α-syn aggregation formation [92]. Exosomes derived from PD patients were found to enhance α-syn accumulation and induce neuronal degeneration in PD mice in a dosage dependent manner. Glial cell-derived exosomes (GDEs) play a crucial role in facilitating communication between neurons and glia. Alpha-synuclein could be transferred from glial cells to neurons via GDEs [93]. Chemokines of GDEs could bind the toll-like receptor 2 (TLR2) and the toll-like receptor 4 (TLR4) of neurons and trigger neuroinflammation and even dopaminergic neuron apoptosis [94]. GDE could also transport aberrantly expressed miRNAs which could trigger or propagate neuroinflammation in neurons [95,96]. In the PD patients’ plasma-derived exosomes, *miR-44438* was significantly upregulated, and inhibited the α-syn efflux from neurons which aggravated α-syn accumulation and aggregation [97]. A few groups have demonstrated that disease-associated miRNAs, such as *miR-19b*, *miR-24*, *miR-195*, and *miR-331-5p*, are remarkably altered in PD patients’ plasma-derived exosomes, and those miRNAs mediated procedures of PD pathogenesis, including inflammation, protein aggregation, and inhibition of autophagy [98,99,100]. 

In summary, exosomes play a crucial role in PD pathogenesis by disseminating α-syn and promoting its aggregation. Moreover, GDEs transfer α-syn or inflammatory factors to neurons, exacerbating the progress of PD. Overall, exosomes are involved in PD pathogenesis in multiple aspects.

### 4.2. Exosomes-Based Diagnosis of PD

Currently, PD is diagnosed mainly based on the presence of obvious clinical motor symptoms such as resting tremor, muscle rigidity, bradykinesia, stooped posture, and the results of bioimaging [93]. The clinical symptoms-based diagnosis was somehow subjective and retarded, and bioimaging was tedious and costly. An alternative diagnostic way was desired, and exosomes emerged as one of the promising candidates.

Researchers reported that α-syn in exosomes derived from PD patients’ body fluids was upregulated compared with that of a healthy control [89]. This group also found that the level of tau, one hyperphosphorylated and aggregated protein in tauopathies, was significantly elevated in exosomes from PD patients’ plasma. The authors speculated that exosomes might serve as the carrier to transport tau from brain to blood and this made it more feasible to detect the level of PD associated tau [101]. Phosphorylated leucine-rich repeat kinase 2 (LRRK2), another PD associated protein, was found up-regulated in the exosomes of PD patients’ urine, and the level of Phospho-LRRK2 was correlated with the impairment of PD patients [102]. Moreover, Wang et al. reported that synaptosome associated protein 23 (SNAP23) and calbindin in exosomes from PD patients’ urine was higher than those of the control [103]. Rennika Kluge et al. reported that a pathological form of α-syn in neuron-derived exosomes of PD patients’ plasma was obviously increased compared with that of healthy controls. The extracted α-syn showed more β-sheet-rich structures and fibrillary appearance which could result in the accumulation of amyloid proteins after amplification [104]. More concretely, Leng et al. analyzed the protein expression profile in neuron-derived exosomes from PD patients with and without rapid eye movement sleep behavior disorder (RBD). They found that exosomes from PD patients with RBD had lower levels of excitatory amino acid transporters-2 (EAAT-2) and vesicular glutamate transporter type 1 (VGLUT-1), suggesting that they could be used as predictors to differentiate subtypes of PD patients [105]. 

MiRNAs are another kind of biomolecule usually present in exosomes. Pallabi Bhattacharyya et al. reported that *miR-128* was down-regulated in plasma-derived exosomes from PD patients [106]. Yao et al. reported that the level of *miR-331-5p* from PD patients’ plasma exosomes was increased, while that of *miR-505* was decreased [107]. Cao et al. found that the level of *miR-24* and *miR-195* in PD patients’ exosomes were elevated, while that of *miR-19b* was reduced [108]. Cheng et al. analyzed the circRNAs of exosomes from a 1-Methyl-4-phenyl-1,2,3,6-tetrahydropyridine (MPTP) induced mice PD model and found that *circSV2b* was downregulated, indicating that *circSV2b* had the potential to serve as a new biomarker in the diagnosis of PD [109]. Most of the diagnostic biomarkers mentioned in Figure 3 have only been validated in animal models of PD. Among them, α-syn has been utilized as a clinical biomarker for PD diagnosis [110]. It is notable how a recent clinical trial (NCT01860118) has validated that LRRK2 derived from exosome could serve as biomarkers for PD.

Collectively, proteins, miRNAs, circRNAs, and other molecules derived from exosomes could serve as biomarkers for PD diagnosis. It is notable how exosomes could be obtained from various body fluids including CSF, plasma, serum, saliva, and urine, which ensure the accessibility and availability of those biomarkers. Therefore, exosomes could serve as valuable resources for PD diagnosis. 

### 4.3. Exosomes-Based PD Therapy

In the clinic, PD patients are treated by administering levodopa, surgery, and deep brain stimulation. Yet the effect of those treatments is unsatisfactory and the side effect is unneglectable. New interventive methods need to be explored (Figure 3). Exosomes attract attention due to their features including good biocompatibility, higher capacity for drug delivery, and penetrating the BBB [111]. 

Jarmalavičiūtė et al. reported that exosomes from stem cells derived from the dental pulp of human exfoliated deciduous teeth (SHEDs) displayed a neuroprotective effect on 6-hydroxy dopamine (6-OHDA) triggered dopaminergic neuron [112]. Narbute et al. proved that intranasal administration of exosomes from SHEDS ameliorated dyskinesia and dopaminergic neuron loss in PD rats [113]. Shakespear et al. reported that an astrocytes-derived exosomes-rescued neurotoxin, N-Methyl-4-phenylpyridinium Iodide (MPP^+^), induced dopaminergic neuron death [114]. Another group reported that exosomes derived from astrocytes stimulated by chemokine CCL3 exhibited remarkable protective ability on H_2_O_2_-induced apoptosis of differentiated SH-SY5Y. Furthermore, exosomes derived from the ventral intraventricular region enhance ATP production and survival in SH-SY5Y subjected to MPP^+^ [115].

Recently, exosomes have been engineered to serve as a platform for PD therapy. Exosomes loaded with peroxidase effectively deliver peroxidase into neurons, and microglia exert a neuroprotective effect in substantia nigra pars compacta (SNpc) neurons in PD mice [116]. Qu et al. demonstrated that exosomes loaded with dopamine after tail vein injection could be delivered into the brains of PD mice, and this approach showed alleviated symptoms. Dopamine-loaded exosomes increased the brain dopamine concentration of a mice PD model by more than 15-fold [117].

In addition, Yang et al. demonstrated that exosomes-mediated delivery of ASO4, one antisense oligonucleotides targeting human α-syn sequence, significantly reduces α-syn expression and aggregation, and ameliorates dopaminergic neuron degeneration in PD mice [118]. All those studies showed that original or engineered exosomes could hold great potential for PD treatment. The level of NDDS related “cargo” in exosomes is summarized and listed in Table 2 [65,66,67,89,102,103,104,107,108,109,119,120,121,122,123,124,125,126].

## 5. Engineered Exosomes and Drug Delivery

Exosomes have exhibited potential for serving as biomarkers, drug delivery agents, and therapeutic tools. Yet there are still certain limitations of native exosomes which might hinder the application of exosomes. For example, the “cargo” of exosomes is limited by its origin. Exosomes from different resources have distinct functions and could not be regarded as possessing uniformity. Native exosomes also lack targeting capabilities, and it is necessary to develop methods for targeted delivery. One more aspect about exosomes to be noted is their short half-life and how they are rapidly cleared by immune cells [127,128]. Hence, there is an urgent need to modify or engineer exosomes so as to endow them with better practicability. One approach is to engineer exosomes to load specific therapeutic “cargo”, such as drugs, RNA molecules, or proteins [129]. It has been reported that exosomes loaded with levodopa improved the motor impairments of PD mice after being administered via tail vein injection. Another strategy is to modify the surfaces of exosomes to improve their stability, targeting capabilities, or immune system evasion, which can be achieved by modifying membrane proteins or introducing specific ligands on the outside edges. Researchers have modified exosomes with RVG peptide and it renders exosomes specifically delivering the “cargo” to neurons in the brain [80]. Through modification or engineering, exosomes can be optimized to enhance their functionality, paving the way for innovative diagnostic approaches and therapeutic interventions in these neurodegenerative disorders.

In the context of AD/PD therapeutics, different delivery methods for exosomes have been proposed. Intranasal delivery offers a non-invasive and convenient approach, allowing direct access to the brain through the olfactory pathway [130]. Intravenous injection provides systemic delivery, although it faces challenges in crossing the BBB. Intracerebroventricular injection or direct brain injection enables specific brain regions delivery, but which is one invasive procedure. Focused ultrasound can temporarily disrupt the BBB so as to facilitate exosome brain delivery [131]. Additionally, exosomes coated with nanoparticles show enhanced BBB penetration and increased payload capacity [132]. It should be noted that the choice of delivery method depends on multiple factors, including the exact therapeutic goals, the nature of the exosomes, the desired target brain region, and safety considerations [133,134]. Each method has its own limitations, and optimizations surrounding these methods are ongoing for more efficient delivery into the brain (Table 3). Another significant aspect to consider is the uptake capacity of target cells. Increasing the cellular uptake of exosomes by target cells may potentially enhance therapeutic efficacy. 

## 6. Summary and Perspectives

Collectively, given their convenient availability, good biocompatibility, and flexibility for engineering, exosomes attract ascending attention for potential biological application in basic research as well as in clinic especially for AD and PD. “Cargo” of exosomes could serve as diagnostic biomarkers. Original or engineered exosomes could be adopted for the clinical treatment of AD and PD. Notably, for the better application of exosomes, some aspects deserve to be considered. Firstly, biomarker detection based only on exosomes is not deterministic for AD and PD diagnosis, and supportive information from manifestations or bioimaging of patients is still needed. Secondly, exosomes from different cells or resources might have distinguished “cargo” functions, and exosomes should not be assumed to be uniformly applicable. Thirdly, exosomes applied in vivo could be transported to the liver and spleen for degradation before they reach the brain to exert their roles, and the ultimate therapeutic effect is debatable. Last but not least, more efforts are still needed to enhance the applicability of exosomes by improving neuron targeting ability, drug-loading capacity, and responsiveness to pathological environments. Exosomes hold great potential in clinical application, which could be strengthened by modification or engineering.

## Figures and Tables

**Figure 1 ijms-24-11054-f001:**
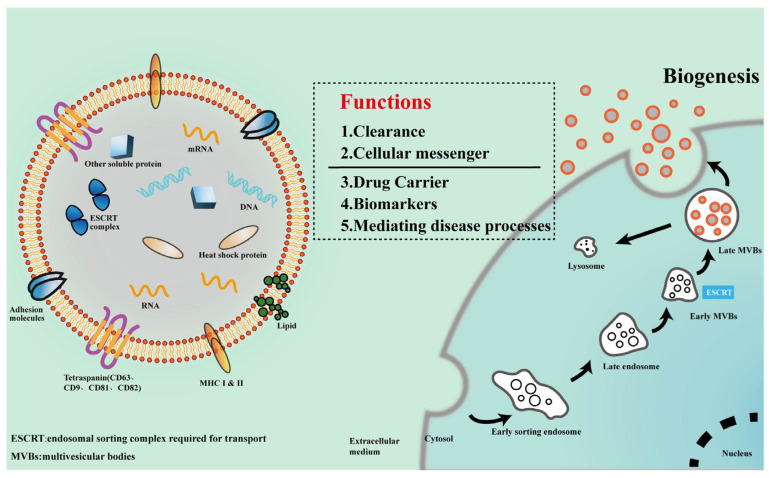
Structure, biogenesis, and function of exosomes. The right panel illustrates the biogenesis process of exosomes. The cell membrane undergoes inward invagination, forming early sorting endosomes, which then mature into late endosomes. With the assistance of ESCRT, “cargo” is loaded into MVBs, leading to the formation of late MVBs. Some exosomes may undergo degradation through lysosomes, while late MVBs can release exosomes into the extracellular space through membrane fusion. The left panel depicts the basic structure of exosomes and their components. The middle section highlights the potential functions that exosomes may exert.

**Figure 2 ijms-24-11054-f002:**
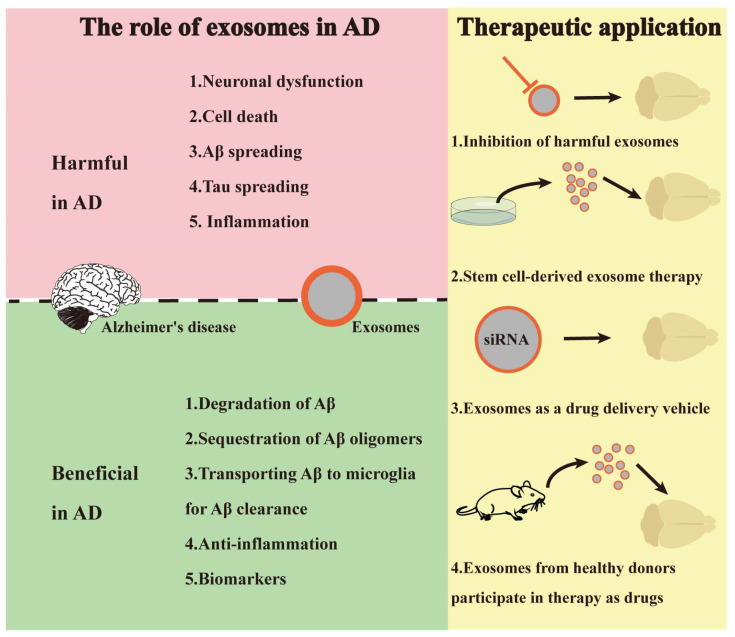
The role of exosomes in AD and exosome-based therapies. The left panel depicts the potential “harmful” or “beneficial” effects of exosomes on AD. Harmful effects include inducing neuronal dysfunction, cell death, contributing to the spread of Aβ/Tau, and triggering inflammation. Conversely, exosomes also have beneficial effects on AD such as participating in Aβ clearance, anti-inflammation, or serving as diagnostic biomarkers. The right panel demonstrates the potential therapeutic applications of exosomes in AD, such as by directly exerting neuroprotective effects through exosomes derived from stem cells or healthy donors, or by serving as carriers for drug delivery.

**Figure 3 ijms-24-11054-f003:**
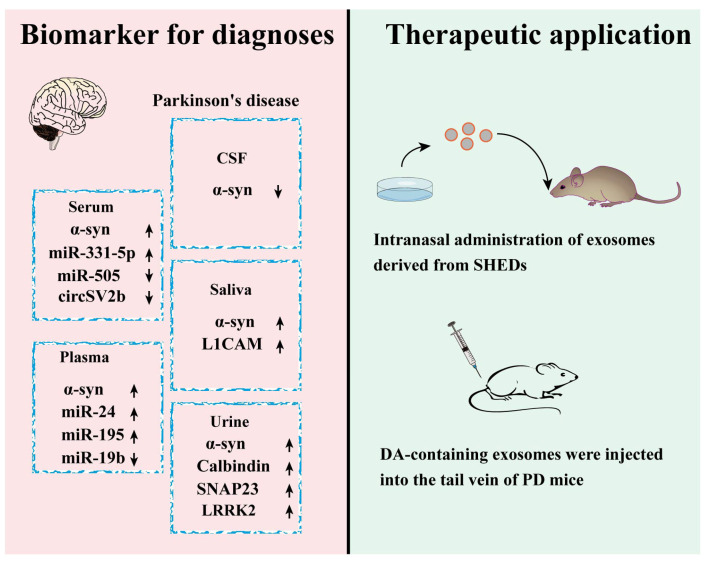
Exosomes-based diagnostics and therapeutics for PD. Cargos of exosomes derived from various bodily fluids such as CSF, serum, plasma, saliva, and urine could serve as PD biomarkers, including α-syn, miRNA, and proteins. The right panel showcases exosome-based therapeutics approaches, such as intranasal delivery of exosomes derived from SHEDs and tail vein injection of engineered DA-loaded exosomes.

**Table 1 ijms-24-11054-t001:** Major types of EVs.

	Markers	^1^ EV Calss	Name	Size
Exosome	CD63 CD81 CD9	^2^ sEV	Exosome (Classical)	40–150 nm
		sEV	Exosome (Non-Classical)	40–150 nm
Microvesicel	Annexin A1 ARRDC1	^3^ lEV	Microvesicel (Classical microvesicle)	~150–1000 nm
		sEV	Microvesicel (ARMM)	~40–100 nm
Apoptotic EV	Annexin V	lEV	Apoptotic EV (Apoptotic body)	1–5 μm
		sEV~lEV	Apoptotic EV (Apoptotic vesicle)	~100–1000 nm
Autophagic EV	LC3B-PE p62	sEV~lEV	Autophagic extracellular vesicle (Autophagic EV)	40–1000 nm
oncosomes	Annexin A1	lEV	oncosomes	1–10 μm

^1^ EV, extracellular vesicle; ^2^ sEV, small EVs are <200 nm in diameter; ^3^ lEV, small EVs are >200 nm in diameter.

**Table 2 ijms-24-11054-t002:** The alterations in the “cargo” of exosomes in AD and PD.

Disease Types	Exosome Sources	Exosomal Biomarker	Level	Ref.
AD	Serum	*miR-135a*	up-regulated	[66,119,120,121,122,123,124]
		*miR-384*		
		*miR-193b*		
		*miR-126-3p*		
		*miR-138-5p*		
		*miR-659-5p*		
		*miR-5001-3p*		
		*miR-361-5p*		
		*miR-30e-5p*		
AD	Serum	*miR-23b-3p*	down-regulated	
		*miR-24-3p*		
		*miR-29b-3p*		
		*miR-125b-5p*		
AD	CSF	*miR-132-5p*	up-regulated	[67]
		*miR-125b-5p*		
		*miR-485-5p*		
AD	CSF	*miR-16-2*	down-regulated	
		*miR-29c*		
		*miR-331-5p*		
AD	blood neuron derived exosome	*miR-212* *miR-132*	down-regulated	[125]
AD	brain tissue	(PE) molecules (p-36:2, p-38:4)	up-regulated	[65]
PD	Serum	α-syn	up-regulated	[89,107,109]
		*miR-331-5p*		
PD	Serum	*miR-505*	down-regulated	
		*circSV2b*		
PD	plasma	α-syn	up-regulated	[104,108]
		*miR-24*		
		*miR-195*		
PD	plasma	*miR-19b*	down-regulated	
PD	saliva	α-syn	up-regulated	[126]
		L1CAM		
PD	urine	calbindin	up-regulated	[102,103]
		SNAP23		
		LRRK2		

**Table 3 ijms-24-11054-t003:** Different methods for exosomes delivery.

Delivery Methods	Advantages	Disadvantages
①Engineered Exosomes	①Precise targeting to receptors on the BBB, increasing the chances of crossing; ②Flexibility to customize exosomes for specific applications	①Complex engineering process that may affect natural characteristics of exosomes; ②Challenges in achieving optimal targeting efficiency and maintaining engineered exosome stability
②Intravenous Injection	Systemic delivery, allowing exosomes to reach various organs including the brain.	①Low efficiency in crossing the blood-brain barrier; ②Exosomes may undergo clearance by the liver and other organs before reaching the brain.
③Intracerebroventricular Injection	①Allows direct and localized delivery of exosomes to specific brain regions; ②Bypasses the blood-brain barrier	Invasive procedure requiring surgical intervention; Limited to targeted brain regions
④Direct Injection into Brain Tissue	Precise delivery to specific brain regions; Allows for localized effects	Invasive procedure requiring surgical intervention; Limited to targeted brain regions
⑤Focused Ultrasound	①Allows temporary and localized opening of the BBB, enabling exosomes to pass through; ②Non-invasive method with potential for delivering various therapeutic agents	①Precise targeting and control of BBB opening required to avoid potential side effects; ②Safety and long-term effects of the method require further investigation
⑥Intranasal Delivery	①Non-invasive and relatively simple method for delivering exosomes to the brain; ②Bypasses the BBB through the olfactory and trigeminal pathways, providing direct transport to the brain	① Amount of exosomes reaching the brain may be limited; ②Distribution of exosomes throughout the brain may not be uniform

## Data Availability

Data sharing is not applicable as no new data were created or analyzed in this study.

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
