# Peer review of "Role of Exosomes in the Pathogenesis and Theranostic of Alzheimer’s Disease and Parkinson’s Disease"

_ijms, 2023, doi:10.3390/ijms241311054_

Round 1

Reviewer 1 Report

Alzheimer's disease (AD) and Parkinson’s disease (PD) is a progressive neurodegenerative disorders (NDs) that causes dementia and eventually death. There is currently no efficient treatment available to slow or stop the progression of AD and PD. In this review, the authors have discussed the role of exosomes. In the present review, authors have summarized and elaborated the properties of exosomes, and their potential application in the theragnostic of AD and PD.

In my opinion, the review is important and timely to address some important factors against neurodegenerative diseases. Overall, I am positive about this manuscript, but before making a favorable decision, I have some concerns and comments about the present form of the manuscript that must be addressed first.

Comments for authors

Comment 1: The abstract primarily provides background information, but it is crucial to highlight the novelty of this review in relation to the existing literature on the subject. I recommend rewriting the abstract.

Comment 2: AD and PD can be caused by a variety of factors. When comparing the number of factors that cause NDs, the authors' description offered in the background (introduction) is insufficient to convey the information, which should be increased for new readers in a review article. The microwave was also thought to be responsible for NDs, especially AD. I encourage authors to add some background on this topic. The suggested article may assist authors in expanding their background knowledge and understanding the mechanisms by which the electromagnetic field interacts with and affects biological systems for various effects.

Article: Microwave Radiation and the Brain: Mechanisms, Current Status, and Future Prospects. International Journal of Molecular Sciences vol. 23 (2022). [https://doi.org/10.3390/ijms23169288].

Comment 3: Exosomes in immune system regulation and immune cell communication need to discuss extensively in the introduction section. Also, what are the mechanisms by which exosomes cross biological barriers, such as the blood-brain barrier?

Comment 4: The revision of Figure 2 is necessary. The label "harmful" in the top left corner is unclear, and it is difficult to discern whether it refers to the harmful effect of exosomes or solely to AD. From my understanding, the authors aim to illustrate the harmful effects of AD alone and then highlight the benefits of exosomes, which are indicated as "beneficial." To avoid misleading readers, it would be beneficial to revise the headings. Additionally, it is crucial to provide clear and precise information that accurately conveys the intended message. I would like to see the revised version of this figure.

Comment 5: Can exosomes be engineered or modified to enhance their therapeutic potential?

Comment 6: Based on the comprehensive literature review, it is essential to incorporate closing remarks and provide a perspective on the subject matter. The conclusion seems to only future perspective. Authors need to conclude from the literature which is summarized in the manuscript.

Comment 7: Can exosomes be engineered or modified to enhance their therapeutic potential? 

Comment 8: I was unable to find a comprehensive discussion on the summarized literature. Therefore, it is necessary to enhance and expand upon the discussion of the reviewed literature. Furthermore, it is important to provide clear explanations of the figures within their respective figure legends.

Comment 9: The paper contains errors and typos that make it difficult to understand and distort its intended meaning. I encourage authors to reread carefully and fix any grammatical errors.

The paper contains errors and typos that make it difficult to understand and distort its intended meaning. I encourage authors to reread carefully and fix any grammatical errors.

Author Response

Thanks for reviewing our manuscript. Your comments and suggestions are constructive. We have revised the manuscript accordingly. For the revision, please refer to the point-by-point response letter.

Reviewer 2 Report

In this review, the authors attempted to provide an overview of the role of exosomes in Parkinson's and Alzheimer's diseases. The authors mentioned the biogenesis of exosomes, the pathogenesis of the disease, and the possible theragnostic potential.

Several sentences with respect to exosomes are repetitive.

It would be helpful to the readers if the authors summarized the exosome-based diagnosis in tables. 

While numerous examples were cited, some citations need further elaboration. For example, in sentence 170, the authors mentioned that microglia-derived exosomes participated in AD pathogenesis. Please elaborate.

The figures are confusing. Figure legends need further explanation. 

Future directions, and limitations of exosomes as biomarkers, delivery agents, and theragnostic are missing. Please also discuss ways to overcome these limitations. 

The manuscript contains numerous typographical and syntax errors.

Author Response

(The authors gave the same response as above.)

Reviewer 3 Report

In this review, the authors address the properties of exosomes and their potential applications in the diagnosis and treatment of AD and PD. The review aims to pave the way for exploring the bioapplications of exosomes in basic research and clinical settings related to NDDs. The authors provide a comprehensive overview of the role exosomes play in the context of AD and PD. They address their involvement in pathogenesis and explain how they can be used as potential biomarkers for early diagnosis and what opportunities they offer for the delivery of therapeutics. They describe the current state of exosome research in relation to these diseases and highlight areas that require further investigation. Overall, the review is a detailed and informative analysis of the potential of exosomes in the diagnostics of AD and PD. However, some sections could be improved to provide more comprehensive coverage. However, certain sections could be improved for a more comprehensive coverage.

The introduction could benefit from a reference to the new classification of MISEV2022 extracellular vesicles and discuss the possibility that exosomes contain organelles such as mitochondria (PMID: 35806411, PMID: 32980480).

The authors should point out the limited diagnostic value of quantitative analysis of exosomes and their size, as such changes have been detected in a variety of pathologies, including those related to age.

The following statement is questionable: "Compared traditional methods, exosomes-based detection is cheaper, simpler, and less invasive." The debatable statement about the advantages of exosomes-based detection over traditional methods could be clarified by mentioning current standard methods for isolation of extracellular vesicles. Could the authors supplement this section with a description of the presence of certain receptors associated with nervous tissue origin on the surface of exosomes?

Based on the studies reviewed, can the authors draw a conclusion about the preferred route of delivery of exosomes to the brain, such as intranasal or intravenous administration?

With these improvements, the review could provide a more solid and in-depth analysis of the potential of exosomes in the diagnostics of AD and PD.

Author Response

(The authors gave the same response as above.)

Round 2

Reviewer 1 Report

The authors have done an excellent job of addressing all of my comments and concerns in the revised version of the manuscript. Furthermore, noticeable improvements were made in the quality of the English language and figures. Given these significant modifications, I firmly believe that the manuscript is deserving of publication in the esteemed journal, IJMS.

Author Response

Thanks for your positive comments and recognization about our revision.

Reviewer 2 Report

Please include more description in the figure legend especially Figure legend 2 and 3. As it is, it is difficult for the readers to understand.

Extensive editing of the English language is required to make the manuscript easily understand.

Please check your spelling.

Line 500 and 506: "contains", do you mean "content"? Please check the manuscript thoroughly.

Sentences in the manuscript read like individual facts strung together with little explanation. It would help the reader to appreciate the review if the authors elaborate and explain their points. For eg. sentence 94: "Moreover, these extracellular vesicles (EVs) have been found to undergo specific changes in their cargo composition in relation to AD and PD." Sentence 206"...depending on the type of microglial cells..." Please check, elaborate and explain.

Are the biomarkers shown in Figure 3 used in the clinics as diagnostic tool for PD? please discuss

Extensive editing of the English language is required to make the manuscript easily understand.

Please check your spelling. 

Author Response

Thanks for your critical comments and suggestions. We have revised the manuscript accordingly. Please refer to the point-by-point response letter.

Round 3

Reviewer 2 Report

The authors have addressed my concerns.

Improved, but minor editing still required